# Flatten the curve: Empirical evidence on how non-pharmaceutical interventions substituted pharmaceutical treatments during COVID-19 pandemic

**Weiyu Luo**[1], **Wei Guo**[2], **Songhua Hu**[1], **Mofeng Yang**[1], **Xinyuan Hu**[1], **Chenfeng Xiong**[1,3]*

1 Maryland Transportation Institute (MTI), Department of Civil and Environmental Engineering, University of Maryland, College Park, MD, United States of America, 2 Asia-Pacific Academy of Economics and Management and Faculty of Business Administration, University of Macau, Macau, China, 3 Shock Trauma and Anesthesiology Research (STAR) Center, School of Medicine, University of Maryland, Baltimore, MD, United States of America

* cxiong@umd.edu

**Data Availability Statement:** The codes and dataset are publicly available on Figshare. The link is https://doi.org/10.6084/m9.figshare.16641358. v1.

## Abstract

During the outbreak of the COVID-19 pandemic, Non-Pharmaceutical and Pharmaceutical treatments were alternative strategies for governments to intervene. Though many of these intervention methods proved to be effective to stop the spread of COVID-19, i.e., lockdown and curfew, they also posed risk to the economy; in such a scenario, an analysis on how to strike a balance becomes urgent. Our research leverages the mobility big data from the University of Maryland COVID-19 Impact Analysis Platform and employs the Generalized Additive Model (GAM), to understand how the social demographic variables, NPTs (Non-Pharmaceutical Treatments) and PTs (Pharmaceutical Treatments) affect the New Death Rate (NDR) at county-level. We also portray the mutual and interactive effects of NPTs and PTs on NDR. Our results show that there exists a specific usage rate of PTs where its marginal effect starts to suppress the NDR growth, and this specific rate can be reduced through implementing the NPTs.

## 1. Introduction

COVID-19 has brought unprecedented global problems. As of June 10, 2020, the rapidly spreading COVID-19 virus has infected 7,492,360 people and claimed 422,150 lives across the world. The first case of COVID-19 in the US was confirmed in Washington State on January 21, 2020. On March 13, 2020, only two days after the World Health Organization (WHO) announced the COVID-19 as a world-wide pandemic, the U.S. government proclaimed a national state of emergency concerning the COVID-19 outbreak [1]. By mid-April 2020, stay-at-home orders were issued across all but 8 states in mid-west region of the US. This indicates that at least 316 million people were being urged to stay home, reduce unnecessary contact, and keep social distance [2]. However, the restrictions did not last long. On April 16, 2020, the

**Funding:** The authors received no specific funding for this work.

**Competing interests:** The authors have declared that no competing interests exist.

guidelines of reopening the nation were released by the White House. In consonance with that, most of the lockdown states began to reopen in some manner. As of May 1, 2020, 18 states had lifted their stay-at-home orders or partially selected some regions or businesses to reopen [3]. Within only 3 months and 6 days, the number of confirmed cases exceeded 1 million on April 27, 2020 [4].

Until the wide availability of a vaccine, traditional interventions such as social distancing, hand sanitizing, wearing masks and utilizing ventilators remain the primary mechanisms to slow the spread of COVID-19. The governments of different countries have promulgated various countermeasures which can be roughly classified into Non-Pharmaceutical Treatments (NPTs) and Pharmaceutical Treatments (PTs). NPTs are actions, apart from getting vaccinated and taking medicine, that people and communities can take to help slow the spread of illnesses like pandemic influenza [5]. NPTs have been proven to be considerable in delaying and containing the spread of the virus [6–12]. Many state governments issued stay-at-home orders, shut down businesses, and limited gatherings to restrict human mobilities. Several research reveal a positive relationship between human mobility and COVID-19 infections [13, 14]. Also, some researchers mentioned that if people actively cooperate and comply with adaptations such as hand washing, masking, and social distancing, the infectivity and impact of the virus can be alleviated [7, 15]. However, moderate interventions such as school closures, self-isolation of symptomatic individuals, or shielding of older people would probably not be sufficient to control the epidemic and to avoid far exceeding available ICU capacity, even using these measures in combination [16]. Travel restriction policy is particularly useful in the early stage of an outbreak when it is confined to a certain area acting as a major source, but it may be less effective once the outbreak is more widespread at a later stage [17]. Geographic Information Systems (GIS) and big data technologies have played an important role in many aspects to fight against the virus, e.g., spatial tracking, prediction of regional transmission, spatial segmentation of the epidemic risk and prevention level and so on [18]. Mobility patterns are found to be strongly correlated with decreased COVID-19 case growth rates for the most affected counties in the USA [19].

From the perspective of PTs, recommendations on managing the Intensive Care Unit (ICU) as well as the infrastructure, supplies, and staff have been proposed to help ICU practitioners, hospital administrators, governments, and policy makers prepare for the upcoming challenge [20]. The World Health Organization (WHO) developed a suite of complementary surge calculators to help governments, partners, and other stakeholders to estimate potential requirements for essential supplies to respond to the current COVID-19 pandemic [21].

Due to the limited time of the pandemic and the availability of data resources and analytical algorithms, several major knowledge gaps exist and are worthy of attention. First, during a second spike in outbreaks without wide availability of vaccines, governments and people become more concerned with finding the balance point for the economy against the pandemic. However, the vast majority of related research only focuses on non-pharmaceutical intervention methods and ignores the pharmaceutical methods, and there is an even greater lack of discussions on how to make a trade-off on both methods. Second, many previous research oversimplified their models; some critical covariates such as socio-demographics, points of interests, and so on are largely ignored [22].

The main contributions of this research can be summarized as follows. First, this research leverages the aggregated data from large amounts of location-based service data which is more informative and accurate than assumptions on disease transmission rate or reproduction rate [23]. Second, most of the previous researchers solely focused on NPTs or PTs, and no one has studied the substitution effect of NPT and PT. This research is also among the first to quantify the relationship between NPTs and PTs, and to visualize their substitutional impact. This is

innovative and timely research that will provide practical information to the medical system. It also provides a good prediction on New Death Rate, New Case Rate, and New Mild Case Rate. Third, this study contributes to broader general interests. Utilizing the daily updated web portal [24] on the pre-trained models mentioned above, we can deliver days- or weeks-ahead predictions to the public.

## 2. Data source

The data source of our study is a comprehensive national human mobility dataset from the University of Maryland COVID-19 Impact Analysis Platform [24, 25]. This platform incorporated over 150 million anonymous individuals monthly active mobile devices to develop the dataset of person movements for the U.S. These data are collected from individual devices including iOS and Android OS. The risk of re-identification is reduced by applying privacy-preserving techniques, such as aggregating to census block group level or county level. More technical details can be found in this paper [13] and the project report [26]. We utilize the data from February 22, 2020 to May 27, 2020 and conduct all the regressions at county level. The NPT and PT metrics are shown in Table 1. We also employ social demographic metrics as covariates in our models to rule out some static impacts. In addition, we also consider the factors referring to time-varying, weekend dummy, governors' approval rates among states, random effects of different states and federal emergency status in our model.

## 3. Variable explanation

The explanations and abbreviations of all variables are listed in Table 1.

## 4. Models

### 4.1. GAM and data preprocessing

This section provides a detailed description of the GAM (Generalized Additive Model) we employed to examine the NPTs v.s. PTs on controlling New Death Rate, New Case Rate and New Mild Case Rate. GAM was originally developed by Hastie & Tibshirani [27] to blend properties of generalized linear models with additive models. Its linear response variables blend in unknown smooth functions of additional independent variables. The inference of these non-linear smooth functions is the highlight of the GAM methods and often draws more research interests. GAM is more flexible on the assumptions for each independent variable. The functions of independent variables can be either a specified parametric form (such as a polynomial), a non-parametric form, or a semi-parametric form, simply as smooth functions.

As previously mentioned, the data starts from Feb. 22, 2020 when there were no cases in most counties. We could simply "one-size-fits-all" the data at a specific date for all counties; however, due to the variation of the first confirmed case among all counties, doing so may exclude significant information from several counties. In order to preserve as much and as precise information as possible and to differentiate the periods before and after the pandemic, we choose the first case occurrence time point of each county as the cutting point and set to zero all independent variables previous to the first case in the panel data.

Typical linear models fail to capture non-linear effects. To address this model limitation, GAM is adopted to incorporate highly complex non-linear relationships with its flexibility on smooth function forms. Moreover, linear predictors can be mixed in with non-linear predictors to enhance the interpretability of the model. In terms of the GAM structure involved in

**Table 1. Explanation of dependent and independent variables.**

| Abbreviation | Explanation | Type | Source |
|---|---|---|---|
| | *Dependent Variables* | | |
| New Death Rate | 7-day moving average of Daily New Covid-19 death number per 1 million people of each state. | - | From JHU repository (https://github.com/CSSEGISandData/COVID-19/tree/master/csse_covid_19_data). |
| New Case Rate | 7-day moving average of Daily New Covid-19 confirmed case number per 1 thousand people of each state. | - | Calculated by MTI based on JHU repository. |
| New Mild Case Rate | 7-day moving average of Daily New Covid-19 mild case number per 1 thousand people of each state. Calculated by (new confirmed case number–new death number of 7 days later)/population*1000 | - | - |
| | *Independent Variables* | | |
| Weekday | 1 for Monday, 7 for Sunday and 2 to 6 are in between. | - | - |
| TI | Time Index. Days offset from 01/01/2020 as day 0. | - | - |
| age60 | 60 years old population percentage | Sociodemographic | Census Bureau |
| Inc | Median income | Sociodemographic | Census Bureau |
| Afr | African Americans population percentage | Sociodemographic | Census Bureau |
| Hisp | Hispanic Americans population percentage | Sociodemographic | Census Bureau |
| Male | Male percentage | Sociodemographic | Census Bureau |
| PD | People per square mile (Population Density) | Sociodemographic | Census Bureau |
| Hot | Number of points of interests for crowd gathering per 1000 people | Sociodemographic | Calculated by MTI |
| HB | Number of staffed hospital beds per 1000 people | Sociodemographic | ESRI: US Hospital Beds Dashboard |
| FE | Federal emergency tag for each day. 0 for not declaring, 1 for declaring. | - | - |
| Appr | Gubernatorial approval ratings | - | https://ballotpedia.org/Gubernatorial_approval_ratings |
| DTest24 | Number of COVID-19 daily tests per 1000 people. The data is lagged for **24 days** and applied to a 7-day moving average. | Pharmaceutical Treatment | https://covidtracking.com/data/api |
| Ctrip24 | Number of trips per 100 people that cross county borders with origin and destination in the same state. The data is lagged for **24 days** and applied to a 7-day moving average. | Non-Pharmaceutical Treatment | Calculated by MTI |
| Strip24 | Number of trips that cross state borders per 100 people. The data is lagged for **24 days** and applied to a 7-day moving average. | Non-Pharmaceutical Treatment | Calculated by MTI |
| ICU24 | Percentage of ICU units occupied with COVID-19 patients. The data is lagged for **24 days** and applied to a 7-day moving average. | Pharmaceutical Treatment | ESRI: US Hospital Beds Dashboard |
| DTest11 | Number of COVID-19 daily tests per 1000 people. The data is lagged for **11 days** and applied to a 7-day moving average. | Pharmaceutical Treatment | https://covidtracking.com/data/api |
| Ctrip11 | Number of trips per 100 people that cross county borders with origin and destination in the same state. The data is lagged for **11 days** and applied to a 7-day moving average. | Non-Pharmaceutical Treatment | Calculated by MTI |
| Strip11 | Number of trips that cross state borders per 100 people. The data is lagged for **11 days** and applied to a 7-day moving average. | Non-Pharmaceutical Treatment | Calculated by MTI |
| ICU11 | Percentage of ICU units occupied with COVID-19 patients. The data is lagged for **11 days** and applied to a 7-day moving average. | Pharmaceutical Treatment | ESRI: US Hospital Beds Dashboard |

this discussion, the functional form is written as below:

$$g(Y_{ti}) = \beta_0 + \sum_{m=1}^{M} \beta_m X_{tim} + \sum_{n=1}^{N} f_n(X_{tin}) + \sum_{q \neq p}^{P} \sum_{p}^{P} f_{pq}(X_{tip}, X_{tiq}) + \epsilon_{ti}$$

In order to smooth out the in-week pattern and some short-term fluctuations, dependent variables and some predictors are calculated by a 7-day moving average. $Y_{ti}$ represents New Death

Rate, New Case Rate, and New Mild Case Rate on day $t$ of county $i$. $g(.)$ represents the link function between independent variables and the dependent variable; Several research assumed that the virus reproduction numbers are gamma distributed in transmission modeling [28–30], here we also assume the dependent variable following a Gamma distribution applying the log link function on the left-hand side of the formula. $X_{tim}$ is the $m^{th}$ fixed effect linear covariate on day $t$ of county $i$ with a coefficient $\beta_m$ in a set of covariates with total number $M$. $X_{tin}$ is the $n^{th}$ non-linear covariate on day $t$ of county $i$ in a set of covariates with total number $N$. $f_n$ represents a smooth function for $X_{tin}$. $f_{pq}(X_{tip}, X_{tiq})$ denotes the smooth interaction of $X_{tip}$ and $X_{tiq}$ with a function form $f_{pq}$. The models use a thin plate regression spline basis for each smooth function [31]. $P$ denotes the set of variables on which we want to observe the effects of interaction.

We utilize variance inflation factor (VIF) to check the multicollinearity for the linear parts of the model. A threshold of 10.0 for each variable is introduced to filter out the highly multicollinear ones.

## 4.2 Summary of variables

A summary of all variables involved in the models is shown in Table 2. Each variable has 301536 observations.

## 4.3. Model selection

Our aim is to fit a statistical model for each county and the whole nation to understand how the social demographic variables, NPTs, and PTs affect the New Death Rate. Using this model,

**Table 2. Summary of variables.**

| Variable | Mean | SD | Median | Min | Max |
|---|---|---|---|---|---|
| | | | *Dependent Variables* | | |
| New Death Rate | 1.25 | 5.41 | 0.00 | 0.00 | 290.92 |
| New Case Rate | 0.03 | 0.13 | 0.00 | 0.00 | 16.86 |
| New Mild Case Rate | 0.03 | 0.14 | 0.00 | -0.13 | 16.88 |
| | | | *Independent Variables* | | |
| TI | 99.5 | 27.7 | 99.5 | 52.0 | 147.0 |
| age60 | 14.9 | 12.8 | 20.0 | 0.0 | 65.0 |
| Inc | 31690.4 | 28024.2 | 40355.0 | 0.0 | 136268.0 |
| Afr | 6.0 | 12.6 | 0.6 | 0.0 | 87.4 |
| Hisp | 5.6 | 11.3 | 1.8 | 0.0 | 99.1 |
| Male | 50.1 | 2.4 | 49.6 | 41.4 | 79.0 |
| PD | 152.6 | 982.5 | 16.0 | 0.0 | 48341.0 |
| Hot | 77.6 | 69.6 | 99.0 | 0.0 | 699.0 |
| HB | 1.9 | 1.6 | 2.5 | 0.0 | 4.7 |
| FE | 0.6 | 0.5 | 1.0 | 0.0 | 1.0 |
| Appr | 30.5 | 25.4 | 43.0 | 0.0 | 73.0 |
| DTest24 | 0.2 | 0.3 | 0.0 | 0.0 | 2.6 |
| Ctrip24 | 90.2 | 38.9 | 90.1 | 0.0 | 335.7 |
| Strip24 | 18.9 | 25.7 | 8.2 | 0.0 | 377.0 |
| ICU24 | 4.2 | 12.0 | 0.0 | 0.0 | 119.5 |
| DTest11 | 0.3 | 0.4 | 0.2 | 0.0 | 2.8 |
| Ctrip11 | 89.9 | 38.7 | 89.7 | 0.0 | 335.7 |
| Strip11 | 18.9 | 25.7 | 8.2 | 0.0 | 377.0 |
| ICU11 | 5.9 | 13.1 | 0.9 | 0.0 | 119.5 |

we predict the future trend of New Death Rate and inspect the relationships between NPTs and PTs. The panel data we leverage is from all counties in the US from Feb. 22, 2020 to May 27, 2020. For a robustness check, another two models fitting New Case Rate and New Mild Case Rate are used. Unequal lag periods are applied to the three models due to different lag time from symptom onset to case confirmation and death. Referring to the previous research [19], an optimal lag of days of 11 is calculated by optimizing the correlation between the pre-defined "mobility ratio" and "COVID-19 growth rate ratio", therefore we chose 11 days for the New Case Rate and New Mild Case Rate models. Referring to the previous research [32], the median time delay is 13 days from illness onset to death. The number "13 days" can also be supported by another previous research [20] which said, "the median time from symptom onset to severe hypoxaemia and ICU admission is approximately 7–12 days". Therefore, we used 11+13 = 24 days as the lag number for the New Death Rate model. The results of the three GAMs are shown in Table 3. The table is divided into two parts: upper half for coefficients corresponding to the linear fixed effects, and the lower half for non-linear smooth terms and interaction terms.

In the lower half, the smooth terms with parameter "bs = 're'" are set to produce a random coefficient for each level of the factor. The abbreviation "e.d.f" is short for "effective degrees of freedom", which reflects the non-linearity of each smooth term. A larger "e.d.f" represents more wiggliness of the smooth term. The adjusted R-sq. is calculated by the formula as follows:

$$Adjusted\ R^2 = 1 - \frac{\sum_i (y_i - f_i)^2 / (n - k)}{\sum_i (y_i - \bar{y})^2 / (n - 1)}$$

The percentage of deviance explained [27] in this paper is calculated as below:

$$\%\ Deviance\ Explained = \frac{Deviance_{Null} - Deviance_{Residual}}{Deviance_{Null}} \times 100\%$$

Where deviance is defined as:

$$D(y, \hat{\mu}) = 2(\log(p(y|\hat{\theta}_s)) - \log(p(y|\hat{\theta}_0)))$$

- Null deviance: $\theta_0$ refers to the null model (i.e., intercept-only model).

- Residual deviance: $\theta_0$ refers to the trained model.

- $y$ represents the outcome.

- $\hat{\mu}$ represents the estimate of the model.

- $\hat{\theta}_s$ and $\hat{\theta}_0$ are the parameters of the fitted saturated and proposed models, respectively. A saturated model has as many parameters as it has training points, that is, $p = n$.

- $p(y|\theta)$ is the likelihood of data given the model.

We've also applied the concurvity function in the "mgcv" package [33] on the three models to measure the concurvity among all the smooth terms in the three models. Generally, a value over 1.0 indicates a term may be a smooth curve of another. In the three GAMs, the concurvity of each combination of two variables among the non-linear parts are less than 1.0, indicating that the concurvity of the formulation is at the acceptable level.

**Table 3. GAM model estimation results (assuming gamma distribution).**

| Dependent Variable | New Death Rate | | | New Case Rate | | | New Mild Case Rate | | |
|---|---|---|---|---|---|---|---|---|---|
| | Parametric coefficients: | | | | | | | | |
| Linear Part | Estimate | P-value | Sig. level | Estimate | P-value | Sig. level | Estimate | P-value | Sig. level |
| (Intercept) | 0.0933 | 0.3000 | | -0.2120 | 0.0000 | *** | -0.2133 | 0.0000 | *** |
| age60 | 0.0042 | 0.0000 | *** | -0.0001 | 0.3499 | | -0.0001 | 0.0789 | . |
| Inc | 0.0000 | 0.0000 | *** | 0.0000 | 0.0000 | *** | 0.0000 | 0.0000 | *** |
| Afr | 0.0207 | 0.0000 | *** | 0.0012 | 0.0000 | *** | 0.0011 | 0.0000 | *** |
| Hisp | 0.0054 | 0.0000 | *** | 0.0013 | 0.0000 | *** | 0.0013 | 0.0000 | *** |
| Male | -0.0082 | 0.0000 | *** | 0.0028 | 0.0000 | *** | 0.0028 | 0.0000 | *** |
| PD | 0.0000 | 0.0000 | *** | 0.0000 | 0.0000 | *** | 0.0000 | 0.0000 | *** |
| Hot | -0.0007 | 0.0000 | *** | -0.0001 | 0.0000 | *** | -0.0001 | 0.0000 | *** |
| HB | 0.0520 | 0.0000 | *** | 0.0143 | 0.0000 | *** | 0.0144 | 0.0000 | *** |
| DTest24 | -0.0572 | 0.0067 | ** | / | / | / | / | / | / |
| Ctrip24 | -0.0009 | 0.0000 | *** | / | / | / | / | / | / |
| Strip24 | 0.0192 | 0.0000 | *** | / | / | / | / | / | / |
| ICU24 | 0.0110 | 0.0000 | *** | / | / | / | / | / | / |
| DTest11 | / | / | / | 0.0158 | 0.0000 | *** | 0.0143 | 0.0000 | *** |
| Ctrip11 | / | / | / | 0.0000 | 0.0001 | *** | 0.0000 | 0.0201 | * |
| Strip11 | / | / | / | 0.0021 | 0.0000 | *** | 0.0020 | 0.0000 | *** |
| ICU11 | / | / | / | 0.0012 | 0.0000 | *** | 0.0012 | 0.0000 | *** |
| | Approximate significance of smooth terms: | | | | | | | | |
| Non-linear Part | e.d.f | P-value | Sig. level | e.d.f | P-value | Sig. level | e.d.f | P-value | Sig. level |
| s(FE, bs = "re") | 0.9843 | 0.0000 | *** | 0.9884 | 0.0000 | *** | 0.9889 | 0.0000 | *** |
| s(Appr) | 7.8948 | 0.0000 | *** | 8.5707 | 0.0000 | *** | 7.8599 | 0.0000 | *** |
| S(Weekday) | 0.1749 | 0.7330 | | 0.0121 | 1.0000 | | 0.0671 | 0.9950 | |
| s(STNAME, bs = "re") | 48.4306 | 0.0000 | *** | 47.6030 | 0.0000 | *** | 47.5256 | 0.0000 | *** |
| s(TI) | 7.8119 | 0.0000 | *** | 7.8680 | 0.0000 | *** | 7.8693 | 0.0000 | *** |
| s(Strip24, ICU24) | 26.1415 | 0.0000 | *** | / | / | / | / | / | / |
| s(Strip11, ICU11) | / | / | / | 26.2568 | 0.0000 | *** | 26.1633 | 0.0000 | *** |
| | Model fit: | | | | | | | | |
| R-sq.(adj) | 0.163 | | | 0.12 | | | 0.108 | | |
| Deviance explained | 47.0% | | | 24.40% | | | 23.0% | | |

Significance codes: 0

'***' 0.001

'**' 0.01

'*' 0.05 '.' 0.1 ' '.

## 5. Results

### 5.1. Inferential analysis

The "R-sq." and "deviance explained" of the New Death Rate model are higher than the New Case Rate and New Mild Case Rate models, reflecting a better explanation of variances and a better improvement from null model to the fitted model for New Death Rate than for New Case Rate from the predictors. Intuitively, the type I and II statistical error of New Case Rate is harder to be reduced than New Death Rate due to the possibility of misdiagnoses, which may result in bias and pollute the data. As for linear effects in the three models, independent variables such as "Population Density" and "Hospital Bed" show the same positive direction of

effects on dependent variables in all three models at a significant level. This indicates a higher risk for people in an area of higher population density and more points of interest. The race factors ("African percentage" and "Hispanic percentage") are fitted as positive impacts on New Death Rate at a significant level. The age factor ("age60") shows significantly positive impacts in the New Death Rate model and insignificant in New Case Rate and New Mild Case Rate model. This indicates that older people are more vulnerable than younger people. With regard to the NPTs ("Cross County trip" and "Cross State trip"), "Cross State trip" significantly influences the dependent variables positively in the three models. However, "Cross County trip" shows relatively weaker linear impacts compared with "Cross State trip" in all three models. As the virus spreads quickly and widely, we continue to observe a positive correlation between PTs ("ICU utilization" and "Daily Test") and dependent variables. This indicates that the pandemic continues to exert great pressure on our medical system throughout the time interval among all counties in our research.

With respect to the non-linear effects in these three models, the estimated degrees of freedom of the significant smooth terms are largely greater than 1.0, indicating a strong non-linearity for the relationship between the smooth terms and dependent variables. The "Weekday" smooth term is insignificant in all three models, indicating there are no weekly patterns of the dependent variables. The P-values of the smooth terms are smaller than 0.001 indicating their statistical significance. Fig 1A–1C show the interactive effect of NPT and PT on the dependent variable in both 3-D and contour plotting. Of particular note is that the New Death Rate decreases when the "Cross State trip" drops. Additionally, where the "Cross State trip" equals 0, the value of effect peaks when "ICU utilization" is approximately 30 and then decreases along the axis. In other words, the marginal effect of "ICU utilization" on "New Death Rate" sharply drops at value 30 where "Cross State trip" equals 0. At the value of "Cross State trip" under 150, the effect remains steady at a lower value. However, as the value of "Cross State trip" becomes larger, a higher utilization rate of ICU is needed to flatten the growth curve of the effect on New Death Rate. For instance, with the value of "Cross State trip" exceeding 150, we may observe a drastic rising effect on "New Death Rate". What stands out in Fig 1A is that the derivative of the effect on New Death Rate reaches 0 at a lower value along the "ICU utilization" axis given a lower "Cross State trip" (the cross-section curve becomes flat) comparing with that given a higher "Cross State trip" value. Given a specific "Cross State trip" value, the point at where the effect of "ICU utilization" reaching its peak and then reducing can be regarded as a "**status shift point**" of our medical system. This peak point gradually disappears along with the increasing of "Cross State trip" value. Therefore, the observation above supports that the "status shift point" is expected to be at an earlier stage along the "ICU utilization" axis when the value of "Cross State trip" decreases. This supports with intuitive thinking that when more people travel in a pandemic, the greater pressure will be imposed on our medical system, and hence make it more challenging for our medical system to subdue the virus. The curved surfaces in Fig 1A–1C indicate quantifiably substitutional impacts of NPT to PT. For example, suppose the "ICU utilization" equals 120. Then, as long as the "Cross State trip" reduces from 225 to 100, the effect on $Ln(New Death Rate+1)$ after 24 days decreases approximately from 2.75 to 1.0 (decrease from 14.64 to 1.72 by converting the effect on $Ln(New Death Rate+1)$ to New Death Rate). Similar findings also appear in the New Case Rate model and the New Mild Case Rate model in Fig 1B and 1C. Same as the New Death Rate model, the more people travel, the "**status shift point**" is expected to be at a later stage or even disappear on "ICU utilization" axis.

## 5.2 Robustness check

To do further robustness checks, we fit 3 new models as comparison by applying an identity link function on dependent variables and assuming them to have a Gaussian distribution. The

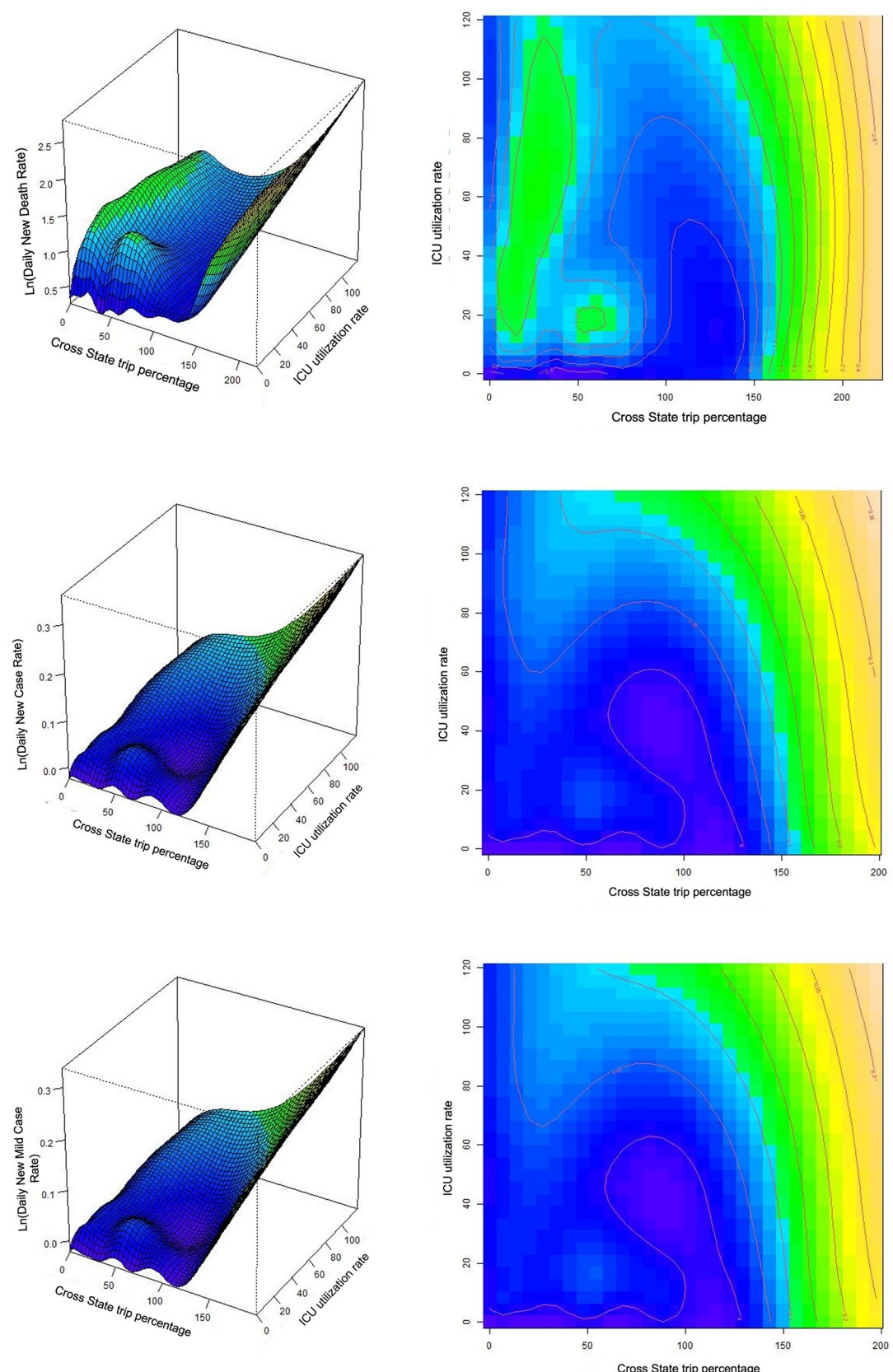

**Fig 1.** (a) Interaction smooth terms of New Death Rate model (Gamma). (b) Interaction smooth terms of New Case Rate model (Gamma). (c) Interaction smooth terms of New Mild Case Rate model (Gamma).

fitting results are listed in Table 4. The findings in the new fitted models are similar to those reported in section 5.1. It shows the robustness on all explanatory variables. Table 5 makes a brief comparison for all 6 models. The adjusted R-sq., deviance explained, AIC and BIC of Gamma models are better than the Gaussian models, showing that the Gamma models fit better than Gaussian models. It indicates that the assumption of Gamma distribution is closer to reality than Gaussian distribution. Comparing the goodness-of-fit among different dependent variables in Gamma models, we observe a better fit for the New Death Rate model than for the New (Mild) Case Rate model. One convincing explanation is that the daily published death number is more reliable than the case number, as case numbers are more likely to be impacted by diverse diagnostic modes, misdiagnosis, restricted test capacity or even political reasons. In contrast, when people die from illness, the related identification documents registered in government agencies have to be modified, which makes the death number more convincing and more difficult to manipulate.

In order to check the robustness of the predictive ability, we also applied the 6 different models to make predictions starting from May 28 to June 10. The training and prediction process is looped weekly. The models are trained by all daily data before a specific day to predict the dependent variable on the next week. The results are shown in Fig 2. The black curves represent the New Death Rate, New Case Rate, and New Mild Case Rate in the subplots, respectively. The green curves stand for models with Gamma distribution assumption. The red curves represent models with Gaussian distribution assumption. The "ICU utilization" is plotted in blue histogram in the background. "Cross State trip" is barplotted in orange as background as well. It is observed that all models show good fittings and predictions.

## 6. Conclusion and discussion

The federal and state governments have enacted various complex combinations of responses to COVID-19. The policies affect various people's activities and result in different patterns of movement and behavioral change. Many state governments have issued mandatory orders impacting economic behaviors, and they have also increased their production and supplies of masks, sanitizers, ventilators and other medical materials. This landscape makes balancing the effect of NPTs and PTs a non-trivial task. This research is based on location-based service data; we captured people's real-time behavior instead of making assumptions of the efficacy of NPT. This means our mobility metrics were driven by the actual number of people who moved across state-bounds or county-bounds.

This research builds models for the whole nation from a macro point of view, treating various states as random effects instead of building a unique model for each state or county. Our results show a strong and statistically significant correlation between New Death Rate, New Case Rate, New Mild Case Rate, and the treatments. We portrayed the variation and captured the "status shift point" of PT at various levels of NPT. Crucially, the quantified interactive and substitutional impact among NPT and PT should serve to support more accurate policy making for state governments to find a better trade-off at an early stage in a pandemic. If cross-state trips are reduced, this would potentially lessen not only new deaths and cases but also new deaths and cases per ICU unit; far less efforts of PTs would be needed to stop the spread of virus; the medical system would operate more smoothly hence the unit efficiency of the medical system would increase.

**Table 4. GAM model estimation results (assuming Gaussian distribution).**

| Dependent Variable | New Death Rate | | | New Case Rate | | | New Mild Case Rate | | |
|---|---|---|---|---|---|---|---|---|---|
| | Parametric coefficients: | | | | | | | | |
| Linear Part | Estimate | P-value | Sig. level | Estimate | P-value | Sig. level | Estimate | P-value | Sig. level |
| (Intercept) | -1.8270 | 0.0000 | *** | -0.2457 | 0.0000 | *** | -0.2457 | 0.0000 | *** |
| age60 | 0.0595 | 0.0000 | *** | 0.0002 | 0.0293 | * | 0.0001 | 0.2549 | |
| Inc | 0.0000 | 0.0000 | *** | 0.0000 | 0.0000 | *** | 0.0000 | 0.0000 | *** |
| Afr | 0.0936 | 0.0000 | *** | 0.0012 | 0.0000 | *** | 0.0011 | 0.0000 | *** |
| Hisp | 0.0313 | 0.0000 | *** | 0.0014 | 0.0000 | *** | 0.0014 | 0.0000 | *** |
| Male | -0.0328 | 0.0000 | *** | 0.0031 | 0.0000 | *** | 0.0032 | 0.0000 | *** |
| PD | 0.0004 | 0.0000 | *** | 0.0000 | 0.0000 | *** | 0.0000 | 0.0000 | *** |
| Hot | -0.0041 | 0.0000 | v | -0.0001 | 0.0000 | *** | -0.0001 | 0.0000 | *** |
| HB | 0.3718 | 0.0000 | *** | 0.0162 | 0.0000 | *** | 0.0161 | 0.0000 | *** |
| DTest24 | 0.5455 | 0.0000 | *** | / | / | / | / | / | / |
| Ctrip24 | -0.0034 | 0.0000 | *** | / | / | / | / | / | / |
| Strip24 | 0.0738 | 0.0000 | *** | / | / | / | / | / | / |
| ICU24 | 0.0413 | 0.0000 | *** | / | / | / | / | / | / |
| DTest11 | / | / | / | 0.0176 | 0.0000 | *** | 0.0160 | 0.0000 | *** |
| Ctrip11 | / | / | / | 0.0000 | 0.0000 | *** | 0.0000 | 0.0031 | ** |
| Strip11 | / | / | / | 0.0025 | 0.0000 | *** | 0.0024 | 0.0000 | *** |
| ICU11 | / | / | / | 0.0013 | 0.0000 | *** | 0.0012 | 0.0000 | *** |
| | Approximate significance of smooth terms: | | | | | | | | |
| Non-linear Part | e.d.f | P-value | Sig. level | e.d.f | P-value | Sig. level | e.d.f | P-value | Sig. level |
| s(FE, bs = "re") | 0.9830 | 0.0000 | *** | 0.9938 | 0.0000 | *** | 0.9933 | 0.0000 | *** |
| s(Appr) | 7.8657 | 0.0000 | *** | 8.4416 | 0.0000 | *** | 7.8780 | 0.0000 | *** |
| S(Weekday) | 0.3003 | 0.5560 | | 0.0487 | 1.0000 | | 0.0050 | 1.0000 | |
| s(STNAME, bs = "re") | 48.7287 | 0.0000 | *** | 47.5349 | 0.0000 | *** | 47.4435 | 0.0000 | *** |
| s(TI) | 7.8136 | 0.0000 | *** | 7.8654 | 0.0000 | *** | 7.8638 | 0.0000 | *** |
| s(Strip24, ICU24) | 26.5259 | 0.0000 | *** | / | / | / | / | / | / |
| s(Strip11, ICU11) | / | / | / | 26.2871 | 0.0000 | *** | 26.1883 | 0.0000 | *** |
| | Model fit: | | | | | | | | |
| R-sq.(adj) | 0.16 | | | 0.116 | | | 0.104 | | |
| Deviance explained | 16.1% | | | 11.60% | | | 10.4% | | |

Significance codes: 0

'***' 0.001

'**' 0.01

'*' 0.05 '.' 0.1 ' ' 1.

**Table 5. Model comparison.**

| Dependent Variable | Assumed Distribution | R-squared | Deviance explained | AIC | BIC |
|---|---|---|---|---|---|
| New Death Rate | Gamma | 0.163 | 47.00% | 1154560 | 1155686 |
| New Death Rate | Gaussian | 0.16 | 16.10% | 1821156 | 1822292 |
| New Case Rate | Gamma | 0.12 | 24.40% | -620096 | -618970.5 |
| New Case Rate | Gaussian | 0.116 | 11.60% | -395662 | -394533.8 |
| New Mild Case Rate | Gamma | 0.108 | 23.00% | -608683.1 | -607568.2 |
| New Mild Case Rate | Gaussian | 0.104 | 10.40% | -380003.7 | -378888.3 |

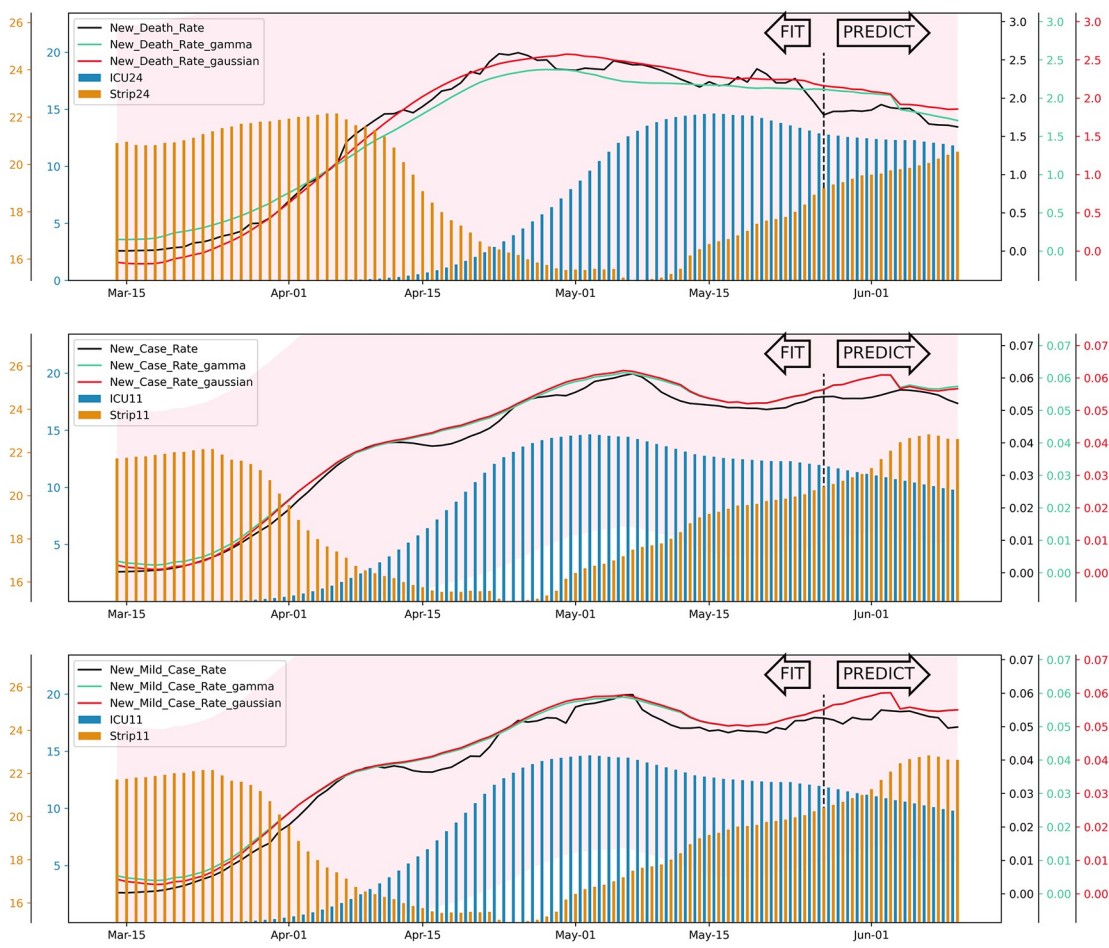

**Fig 2. Fit and prediction of all six models.**

The limitations of this research are summarized below. Though factors regarding spatial density ("PD": Population density, "Hot": Point of Interest) are captured as significant factors, they are both static variables. A combination with travel behaviors and spatial encounters should be involved in a next-step study which reflects dynamic effects for factors referring to spatial density. For further studies aiming at microscopic analysis, it might be possible to utilize zip-code level data or more fine-grained trajectory data to analyze individual encounters and incorporate additional details. Despite this research is mainly focusing on finding the substitution effect, one next step on making better predictions is also significant. Several previous research suggest methods to overcome the model uncertainty and make better predictions. For instance, we might explore applying different types of models and aggregate them by Bayesian Model Averaging methodology [34, 35].

This research mainly discussed a one-way relationship between human mobility and the spread of the COVID-19 virus. However, the opposite direction or the bi-directional relationship also exist between the two variables. Previous research explored the opposite direction on these two variables [11]. For the next step, the authors also plan to apply SEM (Structural Equation Modeling) methodology [36, 37] to explore further on the bi-directional relationship. In addition, people's risk perception over time might significantly affect their behaviors and affect the spread of the virus. In this case, we plan to utilize social media data such as Facebook,

Twitter, Instagram and so on as input to our model to capture people's changing in the risk perception. Several studies have utilized social media data to extract people's perception on COVID-19 related incidents like travelling [38], lockdown [39] and risk mitigation strategies [40].

As local governments will have to lift part of the mandatory orders as the time goes by, more refined analysis on balancing the NPTs and PTs are urgently needed. Our findings highlight a difficult point of decision-making in a pandemic, and also provide a macro view model to support decision making. We hope our study may motivate both individuals and governors to make more optimized decisions to help slow down the spread of pandemic.

## Acknowledgments

We would like to thank and acknowledge Amazon Web Service and its Senior Solutions Architect, Jianjun Xu, for providing cloud computing and technical support.

## Author Contributions

**Conceptualization:** Chenfeng Xiong.

**Data curation:** Weiyu Luo, Songhua Hu, Mofeng Yang.

**Formal analysis:** Weiyu Luo.

**Investigation:** Weiyu Luo, Songhua Hu, Xinyuan Hu, Chenfeng Xiong.

**Methodology:** Wei Guo, Mofeng Yang, Chenfeng Xiong.

**Project administration:** Chenfeng Xiong.

**Supervision:** Chenfeng Xiong.

**Validation:** Mofeng Yang, Xinyuan Hu.

**Visualization:** Mofeng Yang.

**Writing – original draft:** Weiyu Luo, Songhua Hu, Xinyuan Hu, Chenfeng Xiong.

**Writing – review & editing:** Xinyuan Hu.

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
