## [Decision Letter · Decision Letter 0]

2 Aug 2021

PONE-D-21-21763

Flatten the Curve: Empirical Evidence on How Non-Pharmaceutical Interventions Substituted Pharmaceutical Treatments during COVID-19 Pandemic

PLOS ONE

Dear Dr. Xiong,

Thank you for submitting your manuscript to PLOS ONE. After careful consideration, we feel that it has merit but does not fully meet PLOS ONE’s publication criteria as it currently stands. Therefore, we invite you to submit a revised version of the manuscript that addresses the points raised during the review process.

We look forward to receiving your revised manuscript.

Kind regards,

Yajie Zou

Academic Editor

PLOS ONE

Journal Requirements:

 [NA]. 

d) If you did not receive any funding for this study, please state: “The authors received no specific funding for this work.

[We would like to thank and acknowledge our partners and data sources in this effort: (1) Amazon Web Service and its Senior Solutions Architect, Jianjun Xu, for providing cloud computing and technical support; (2) computational algorithms developed and validated in a previous USDOT Federal Highway Administration’s Exploratory Advanced Research Program project; (3) mobile device location data provider partners; and (4) partial financial support from the U.S. Department of Transportation’s Bureau of Transportation Statistics.]

 [NA]

[NA]. 

5. Please ensure that you refer to Figure 1 in your text as, if accepted, production will need this reference to link the reader to the figure.

6. We note you have included a table to which you do not refer in the text of your manuscript. Please ensure that you refer to Table 5 in your text; if accepted, production will need this reference to link the reader to the Table.

Reviewers' comments:

Reviewer's Responses to Questions

**Comments to the Author**

1. Is the manuscript technically sound, and do the data support the conclusions?

Reviewer #1: Yes

Reviewer #2: Partly

2. Has the statistical analysis been performed appropriately and rigorously? 

Reviewer #1: Yes

Reviewer #2: No

3. Have the authors made all data underlying the findings in their manuscript fully available?

Reviewer #1: Yes

Reviewer #2: Yes

4. Is the manuscript presented in an intelligible fashion and written in standard English?

Reviewer #1: Yes

Reviewer #2: Yes

5. Review Comments to the Author

Reviewer #1: This is an interesting paper. Here are some comments from the reviewer.

- There are some other regression models, why choose the GAM model in the analysis?

- Some variables in Tables 3 and 5 are insignificant.

- Compared to the existing literature, what are the novel contributions from this study?

- This study considers many different explanatory variables. One approach to overcome the model uncertainty is the Bayesian model averaging model and this approach can improve the model prediction. The authors are suggested to review and discuss this approach in the study. For example, see: Application of the bayesian model averaging in analyzing freeway traffic incident clearance time for emergency management. Journal of advanced transportation, 2021. Housing appraisal under model uncertainty: Bayesian model averaging method. Advanced Engineering Journal, 1(1), 26-34.

Reviewer #2: Thank you for giving me an opportunity to review this manuscript. This manuscript examines the effect of non-pharmaceutical and pharmaceutical treatments on the new death rates by using the generalized additive model and the state-level data. The reviewer argues that several critical points should be further clarified. Therefore, the reviewer recommends the major revision of the manuscript.

1. Data source: The reviewer suggests the authors provide a more in-depth discussion on their dataset. Although the reviewer acknowledges that the dataset from MTI has been used widely, more description of the data process method and limitation of this dataset should be discussed. For example, does the MTI dataset include location obtained from cell phone trajectory?

2. Table 1: Please consider providing sources of all variables.

3. Model: The reviewer challenges the validity of using the state as a unit of analysis. First, people’s COVID-19 relevant behavior (e.g., perceived perception, travel behavior, wearing masks, and so on) might vary considerably within each state. For example, in many Midwest states, we have observed huge differences between urban and rural residents. Also, given the fact that people’s response to COVID-19 is highly politicized in the U.S., using the state as the unit of analysis cannot capture this important aspect successfully.

4. Model: Moreover, the reviewer understands the potential issue of inter-state trips of the D.C. metro area. This inter-state trip is also important for other parts of the U.S. (e.g., Chicago MSA: IL-IN-WI, Cincinnati MSA: OH-KY-IN). Please provide more details about how the authors address this issue. Also, provide more details about how Figure 1 is created (e.g., data source). Figure 1 has not been addressed in the manuscript.

5. Page 5: The manuscript assumes that “the dependent variable follows a Gamma distribution.” Could the authors please provide scientific references or justification for this assumption?

6. Page 6: Could the authors please provide more justification for choosing 21 and 14 days for different models?

7. Table 1: Some technical terms in the abbreviated form need a brief explanation (e.g., e.d.f.? s(F.E., bs=”re”), and so on).

8. Page 8: The manuscript states that “The R-sq. of the New Death Rate model is remarkably higher than the New Case Rate […], reflecting the data for New Death Rate is more trustworthy than for New Case Rate.” The reviewer challenges this argument because the model fit indices do tell less thing about data quality. In other words, data trustworthiness should not be judged by the model fit indices.

9. Page 12: The results illustrated in Figure 4 are interesting. However, the reviewer wonders why the authors particularly focused on this time of period to check their predictability of the model? Have the authors considered applying the model to the most recent timeframe (e.g., September 2020 – January 2021) to check whether the model also works well?

10. After reading the entire manuscript, the reviewer hopes to ask one important question: How did (or will plan to) the authors address the bi-directional relationship between human mobility and the spread of the COVID-19 virus. This manuscript only considers a one-way directional influence of human mobility on the COVID-19 (e.g., new cases). However, in reality, this would be more of a complex relationship that affects each other. Also, people’s changing risk perception (over time) might significantly affect their behaviors and eventually affect the number of new COVID-19 cases. Could the authors please discuss this issue?

6. PLOS authors have the option to publish the peer review history of their article (what does this mean?). If published, this will include your full peer review and any attached files.

Reviewer #1: No

Reviewer #2: No

---

## [Author Response · Author response to Decision Letter 0]

20 Sep 2021

We have uploaded a separate review response document. Please refer to that document for a detailed list of item-by-item replies.

---

## [Decision Letter · Decision Letter 1]

27 Sep 2021

Flatten the Curve: Empirical Evidence on How Non-Pharmaceutical Interventions Substituted Pharmaceutical Treatments during COVID-19 Pandemic

PONE-D-21-21763R1

Dear Dr. Xiong,

We’re pleased to inform you that your manuscript has been judged scientifically suitable for publication and will be formally accepted for publication once it meets all outstanding technical requirements.

Kind regards,

Yajie Zou

Academic Editor

PLOS ONE

Reviewers' comments:

Reviewer's Responses to Questions

**Comments to the Author**

1. If the authors have adequately addressed your comments raised in a previous round of review and you feel that this manuscript is now acceptable for publication, you may indicate that here to bypass the “Comments to the Author” section, enter your conflict of interest statement in the “Confidential to Editor” section, and submit your "Accept" recommendation.

Reviewer #1: All comments have been addressed

Reviewer #2: All comments have been addressed

2. Is the manuscript technically sound, and do the data support the conclusions?

Reviewer #1: Yes

Reviewer #2: Yes

3. Has the statistical analysis been performed appropriately and rigorously? 

Reviewer #1: Yes

Reviewer #2: Yes

4. Have the authors made all data underlying the findings in their manuscript fully available?

Reviewer #1: Yes

Reviewer #2: Yes

5. Is the manuscript presented in an intelligible fashion and written in standard English?

Reviewer #1: Yes

Reviewer #2: Yes

6. Review Comments to the Author

Reviewer #1: From my point of view, the submission has been greatly improved and thus it is worthy of publication.

Reviewer #2: Thank you for the opportunity to read and comment on the revised submission. I have read the revised paper and paid close attention to the authors' responses to my prior comments and those of fellow reviewers. Overall, this revision addresses the comments made about the shortcomings of the original submission. I greatly appreciate the responsiveness of the authors to my prior comments. I hope the authors stay healthy and safe during this pandemic.

7. PLOS authors have the option to publish the peer review history of their article (what does this mean?). If published, this will include your full peer review and any attached files.

Reviewer #1: No

Reviewer #2: No

---

## [Editor Report · Acceptance letter]

1 Oct 2021

PONE-D-21-21763R1 

Flatten the Curve: Empirical Evidence on How Non-Pharmaceutical Interventions Substituted Pharmaceutical Treatments during COVID-19 Pandemic 

Dear Dr. Xiong:

I'm pleased to inform you that your manuscript has been deemed suitable for publication in PLOS ONE. Congratulations! Your manuscript is now with our production department. 

Kind regards, 

on behalf of

Dr. Yajie Zou 

Academic Editor

PLOS ONE